# From Domestic Sewage to Potable Water Quality: New Approach in Organic Matter Removal Using Natural Treatment Systems for Wastewater

Wojciech Halicki [1,2,*] and Michał Halicki [3]

1   Institute of Applied Ecology, Skórzyn 44a, 66-614 Maszewo, Poland
2   Centre for East European Studies, Faculty of Oriental Studies University of Warsaw, ul. Krakowskie Przedmieście 26/28, 00-927 Warszawa, Poland
3   Faculty of Earth Sciences and Environmental Management, University of Wrocław, pl. Uniwersytecki 1, 50-137 Wrocław, Poland; michal.halicki2@uwr.edu.pl
*   Correspondence: w.halicki.ies@gmail.com

**Abstract:** Natural treatment systems for wastewater (NTSW) allow us to not only reduce environmental pollution with sewage, but also to facilitate the reuse of water. This study presents almost 2.5 years of operation of a NTSW pilot plant, where the purpose of which was to purify domestic sewage from the building of the Institute of Applied Ecology (with three permanent residents and up to five employees) to the quality of drinking water. The NTSW consists of a septic tank, compost beds, and denitrification, phosphorus, and active carbon beds. With an active area of 3 m$^2$ per person and a hydraulic residence time (HRT) of 6 days (excluding the HRT of the tank of 8 days), the NTSW allowed for a mean reduction of 99%, 95%, and 98% for the biological oxygen demand (BOD), chemical oxygen demand (COD), and total suspended solids (TSSs), respectively. The renewed water was characterized by average concentrations of 2.2 mg $O_2$/dm$^3$, 17.8 mg $O_2$/dm$^3$, 2.1 mg/dm$^3$, 4.9 mg $O_2$/dm$^3$, and 0.6 nephelometric turbidity units for BOD, COD, TSS, oxidation, and turbidity, respectively. Thus, it met Polish and European drinking water requirements in terms of oxidation and turbidity. This water can be reused for toilet flushing and irrigation.

**Keywords:** constructed wetland; water renewal; water reuse; wastewater treatment





## 1. Introduction

Due to the deepening global water deficit and the increasing need for more sustainable water management [1], the reuse of water recovered from wastewater after its treatment and appropriate preparation (e.g., through additional processes such as filtration, disinfection) is gaining importance. In addition, the growing importance is also driven by the continuous development of new and increasingly effective natural technologies that allow for the treatment of wastewater to the quality of clean water [2,3]. One of the directions in the development of new solutions are natural water purification technologies, which include constructed wetland (CW), various types of wastewater pond systems, and combinations of these treatment plants. Further, recently, there has been a significant increase in the number of scientific publications related to constructed wetlands, water reuse, and water renewal. The number of articles available in the Science Direct database (https://www.sciencedirect.com/, accessed on 10 March 2022) related with the "constructed wetland", "water reuse" and "water renewal" keywords increased significantly over the past two decades, amounting to 324, 1889, and 706 in 2000 and 3139, 23,228, and 4129 in 2020, respectively.

So far, the use of water recovered from sewage has focused mainly on its use for irrigation in agriculture. An example of this is the recently introduced Regulation of the European Parliament on the minimum requirements for the reuse of water [4]. The

requirements in this document only apply to the use of water in agriculture. The country with the greatest achievements in the field of reuse of water from wastewater is Israel, which uses 85% of treated wastewater mainly as an additional source of irrigation water in agriculture [5]. In the U.S., by contrast, water reuse varies more from state to state. In California, which is the leading state in this respect, 46% of the water recovered from wastewater is used for agricultural purposes, whereas the remaining 54% is used for municipal, industrial, and natural purposes [6]. In Spain, which, in Europe, is the leader in this respect, approximately 11% of treated wastewater is reused, 62% of the reused water is directed to irrigation in agriculture [7].

The presented situation of water reused from sewage mainly concerns large municipal sewage treatment plants, from which, the outflow, often without additional cleaning processes, is directed to irrigation. On the other hand, where better quality water is required, such as the irrigation of crops or watering urban greenery, the outflow from the sewage treatment plant is subjected to additional processes, such as aeration, coagulation, filtration, and disinfection. In this case, the use of additional water purification processes significantly increases the costs of producing good quality water [8]. This fact is the basic factor limiting the reuse of water recovered from sewage, especially concerning water of very good quality. Moreover, the additional increase in energy consumption, which is necessary to carry out these processes in small-scale treatment plants, may mean that the reuse of water results in more losses than benefits for the environment.

New possibilities for the reuse of water recovered from wastewater are offered by the use of natural treatment systems for wastewater (NTSW). Such treatment systems consume little energy (and often even run without energy) and allow for the reuse of good-quality water recovered from wastewater. Moreover, they are characterized by many additional environmental functions, such as: biomass production, air purification, the improvement of local microclimate parameters, and biodiversity [9]. They can also be used for recreational purposes. Over the past decades, NTSW has shown a high efficiency in the treatment of domestic, agricultural, and industrial wastewater [10–15]. Based on the advantages of the NTSW, the improved wetland system was developed and implemented in Poland at the beginning of the 21st century. In the period from 2000 to 2015, approximately 5600 plants of this type were built in Poland, where, in each of them, treated water is reused for many natural purposes, such as creating new habitats for local fauna and flora, irrigation, or increasing local land retention [16].

Another step further in the water treatment and reuse process could be to recycle the water back into the building and use it to flush toilets. This study aims to describe the performance of a NTSW pilot plant operating by the building of the Institute of Applied Ecology (Instytut Ekologii Stosowanej–IES) in west Poland. As there are currently no uniform requirements regarding the quality of water intended for flushing toilets in Poland and other European Union (EU) countries, we decided that the newly developed and implemented technology must purify water to such an extent that the outflow partially meets the criteria of drinking water quality. The development and effective implementation of such solutions using water recovered from wastewater for flushing toilets and watering would allow for 50% savings of tap water per year consumed by households. In this study, we evaluated the removal efficiency of the NTSW plant over an almost 2.5-year continuous operation. The evaluation is conducted on a basis of the following parameters: biological oxygen demand (BOD), chemical oxygen demand (COD), total suspended solids (TSS), oxidation, and turbidity.

## 2. Materials and Methods

### 2.1. Characteristics of the NTSW Pilot Plant

The NTSW pilot plant was designed and built to treat wastewater discharged from the IES building in Skórzyn, Western Poland (http://ies.zgora.pl/en/home/, accessed on 10 March 2022). A scheme of the NTSW pilot plant is shown in Figure 1. Additionally, Figure 2 presents photographs of the plant. The installation consists of a septic tank, compost beds,

denitrification beds, phosphorus elimination beds, active carbon bed, and a renewed water reservoir. The denitrification, phosphorus, and active carbon beds will hereinafter be referred to as water renewal beds.

- The septic tank has a capacity of 3 m$^3$, and it receives raw sewage from the IES building. Mechanical and partly biological wastewater treatment takes place here.
- The compost bed has an area of 3 m$^2$. This bed is a key element in the process of removing organic matter from wastewater. It consists of three separate beds, each of them with a surface area of 1 m$^2$, working under a different daily hydraulic load, namely 70, 100, and 130 l/m$^2$ for the A, B, and C beds, respectively. The three beds have the same composition and structure. From the top, the beds consist of: an 80 cm layer of compost, a 10 cm layer of coarse sand (granulation from 0.2 to 2 mm), and a 10 cm layer of gravel (granulation from 4 to 16 mm). The compost consists of a mixture of wood chips (80%) and peat soil (20%). The entire compost bed was placed in a greenhouse to reduce the emission of odors and to reduce the negative impact of low temperatures in winter.
- The denitrification bed is used to remove nitrates from wastewater treated in the compost bed. In addition, in this case, there were three beds with an area of 1 m$^2$, each of which was supplied with sewage treated in the corresponding part (A, B, or C) of the compost bed. The beds are filled from the top with a 40 cm layer of swamp sediments and are overgrown with wetland vegetation. Under the sediment layer, there is a 30 cm layer of coarse sand (granulation from 0.2 to 2 mm) and then a 10 cm layer of gravel (granulation from 4 to 16 mm). The bed is not involved in the removal of organic matter contained in the treated water. On the contrary, the mineralization of the sludge in the denitrification process causes a slight increase in the concentration of organic matter in the outflow.
- The phosphorus elimination bed is designed to remove phosphorus compounds from treated wastewater. In addition, in this case, there were three beds with an area of 1 m$^2$ each, and they were supplied with outflows from the corresponding (A, B, or C) denitrification beds. The beds are filled from the top with a 50 cm layer of coarse sand (granulation from 0.2 to 2 mm) mixed with building lime (25 kg). Then, there is a 10 cm layer of gravel (granulation 4 to 16 mm). A slight elimination of organic matter occurs in the phosphorus deposits. The use of building lime to remove phosphorus contributes to an increase in the pH of the renewed water. As a result, the pH of the renewed water ultimately ranges from 9 to 10.5 and shows a constant slight downward trend.
- The active carbon bed is designed to remove organic matter remaining in the treated water. In addition, in this case, there were three beds with an area of 1 m$^2$ each, and they were supplied with outflows from the corresponding (A, B, or C) phosphorus elimination beds. Each bed is filled from the top with a 50 cm layer of coarse sand (granulation from 0.2 to 2 mm) mixed with activated carbon (25 kg). Then, there is a 10 cm layer of gravel (granulation 4 to 16 mm).
- The renewed water reservoir (with a capacity of 5 m$^3$) serves as a store of water recovered from sewage.

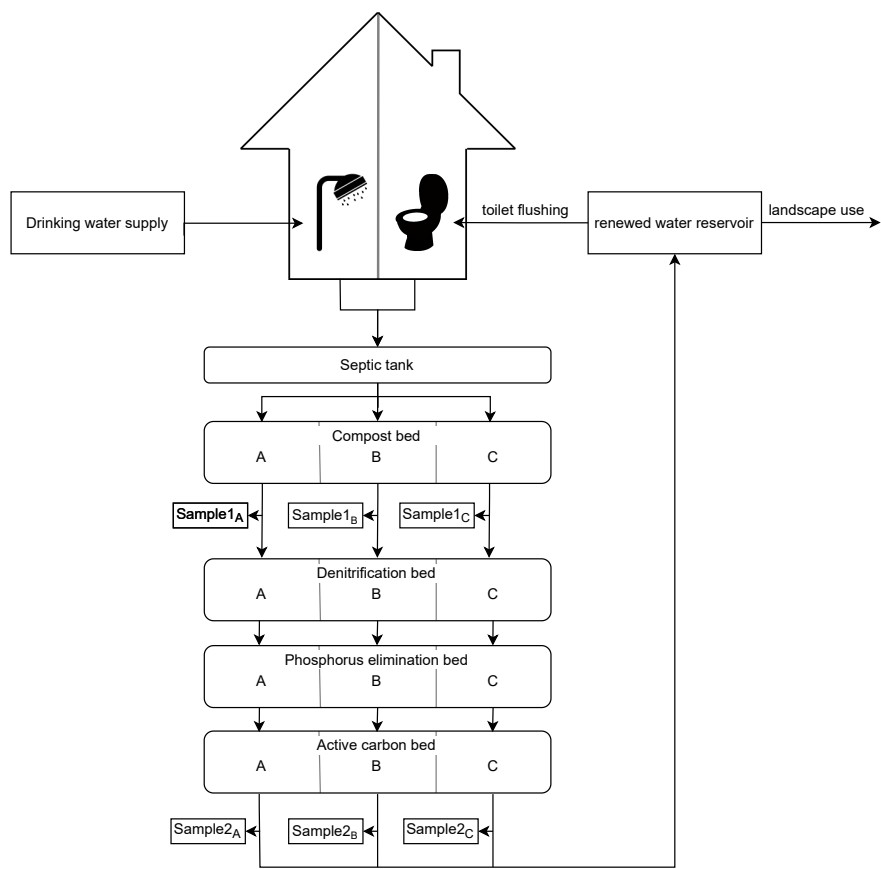

**Figure 1.** Scheme of the pilot plant. The process is conducted under three different daily hydraulic loads: A for 70, B for 100, and C for 130 l/m$^2$.

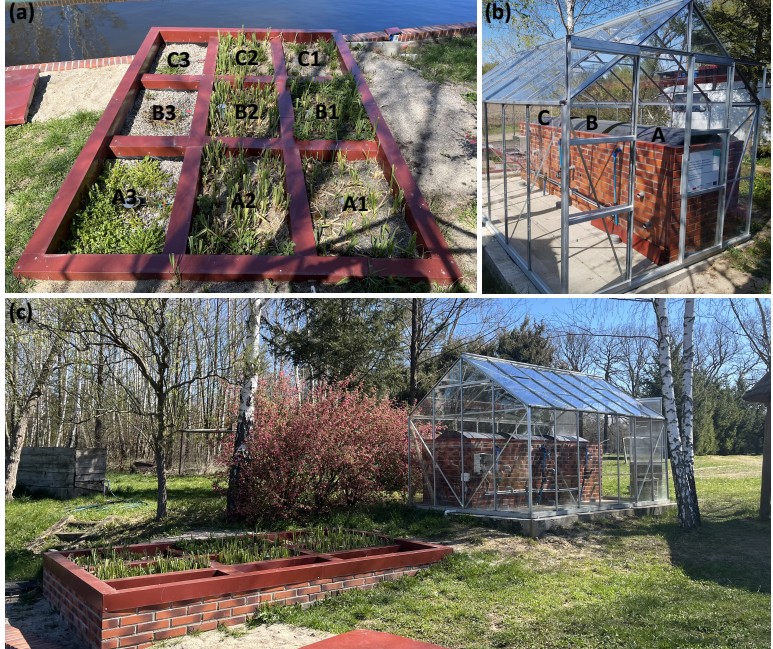

**Figure 2.** The pilot plant. A, B, C letters refer to the three different hydraulic loads (70, 100, and 130 l per day, respectively). (**a**) denitrification (1), phosphorus (2), and activated carbon (3) beds responsible for the water renewal, (**b**) compost beds responsible for the wastewater purification, (**c**) setting of the beds.

### 2.2. Water Sampling and Quality Assessment

The water samples both after the compost beds (Sample 1) and after the water renewal beds (Sample 2) were taken every 2 weeks. Raw sewage (after the septic tank) was sampled once in a month. The analyzes were performed according to Standards Methods for the Examination of Water and Wastewater [17]. The numbers of the following methods are given in brackets. The determination of BOD in the Sample 1 and Sample 2 was performed with the 5-Day BOD Test (5210 B), while in the raw sewage BOD was determined with the Respirometric Method (5210 D). The COD analysis was performed with the Closed Reflux, Colorimetric Method (5220 D). The determination of the total suspension was carried out with the Total Suspended Solids Dried at 103–105 °C Method (2540). Finally, the oxidation and turbidity of the renewed water were determined with the use of the Permanganate Titration Methods and the Turbidity/Nephelometric Method (2130), respectively. As part of the research, organic matter was determined in the renewed water by means of two chemical indicators, i.e. COD and oxidation. The difference between them is that practically all oxidizable organic substances and inorganic substances are oxidized in the determination of COD. It gives a full picture of the actual oxygen demand in the tested water and therefore it is used to determine the oxygen demand in all types of water and wastewater. Oxidation is not as accurate as COD as only about 70% of the organic substance is oxidized during the determination. In practice, it has been assumed that the content of organic matter in drinking water is determined using this method. Therefore, it was assumed that in the framework of this research, the determination of oxygen demand would be carried out using both COD and oxidation in order to have a direct comparison with the requirements that are used in drinking water.

### 2.3. Water Purification Process

The process of wastewater treatment, including the removal of organic matter, begins in the septic tank. Mechanical and partly biological wastewater treatment takes place here. The inflow of sewage to the tank is irregular in time and quantity. It receives sewage from three permanent residents of the IES building and sewage from up to five employees staying at the IES from Monday to Friday for an average of 8 h. On the other hand, the outflow of sewage from the septic tank to the compost beds takes place cyclically every hour, and amounts to a total of 300 l per day. The excess of sewage that arises during the week is collected in the septic tank, and, over the weekend (when there are no employees in the building), it supplies the pilot plant. If, on the other hand, there is an excess of sewage in the septic tank, the load on the compost beds is temporarily increased in order to let the excess sewage pass through the installation. The average hydraulic residence time (HRT) in the septic tank is 8 days.

The main phase of removing organic matter takes place in the compost beds. Wastewater is distributed over the surface of the beds regularly every hour and it slowly seeps through the beds vertically downwards. The average HRT in the compost beds is approximately 1 h. Bacteria and protozoa are essentially involved in the traditional purification processes. Inside the compost beds, a rich environment of typical soil fauna and flora is being created [18]. In the purification process, the compost beds involve heterotrophic and autotrophic bacteria, actinomycetes, fungi, protozoa, nematodes, vases, earthworms, and others [19]. In addition, compost has favorable sorption properties, which contributes to the retention of organic matter in the beds and facilitates their mineralization. After flowing through the compost layer, the treated sewage soaks through the sand layer, in which, above all, the fine suspension washed out of the compost is being removed. The gravel layer enables the outflow of treated sewage from the beds to the next objects. Apart from the removal of organic matter, the process of ammonium nitrogen nitrification also takes place in the compost beds.

The water renewal process takes place in the denitrification, phosphorus and active carbon beds. The HRT of wastewater treated in the each bed is 2 days, which in total gives HRT of 6 days for all three beds at the water renewal stage. However, the first two

beds do not play a significant role in the elimination of organic matter. Nevertheless, the water flows vertically through the denitrification bed, which is under anoxic conditions. These, on the other hand, favor the so-called facultative bacteria, which, under these conditions, reduce nitrates to nitrites and further to gaseous nitrogen, which is released into the atmosphere. As it flows through this bed, water is in direct contact with the organic matter in it, which undergoes constant, slow mineralization. Some of its products dissolve in the flowing water, which sometimes causes an increase in the concentration of BOD and COD. Especially in the first months of operation of the plant, the outflow from the beds was dark due to the excess of rinsed humic substances. After the water flows through the denitrification bed, it then flows by gravity through the phosphorus bed. During this time, a partial reduction in the organic matter remaining in the water takes place. In general, the remaining organic matter in the water is very difficult to decompose, hence its low elimination in the phosphorus bed.

The final stage of the water renewal process is the flow of water through the active carbon bed. As mentioned earlier, the water from the denitrification and phosphorus beds contains organic substance that is difficult to decompose. Therefore, to remove it, a sand bed mixed with active carbon was used. The use of active carbon increases the sorption capacity of the bed. As a result, organic matter that is difficult to decompose is absorbed on the activated carbon and slowly decayed by bacteria. Such treated water (devoid of nitrogen, phosphorus, and organic substances) with the parameters of drinking water is directed to a storage tank, from which, it is then pumped to flush toilets for irrigation and other purposes.

## 3. Results

### 3.1. Wastewater Treatment

3.1.1. Organic Pollutants Removal in the Septic Tank

Raw sewage undergoes preliminary treatment in the septic tank. The average daily amount of wastewater flowing into the tank in the analyzed period, determined on the basis of water consumption, was 340 l on working days and 200 l at weekends. This amount (including three people permanently residing in the IES building and a maximum of five office workers) is consistent with the norms of daily water consumption in Poland of 80–100 l per inhabitant and 15 l per office worker [20]. The numbers of daily loads of pollutants assumed per person are 60, 120, and 55 g for BOD, COD, and TSS, respectively [21]. A quarter of these values were assumed for office workers. It can therefore be assumed that the pilot plant treated wastewater from four permanent residents. Based on the assumptions presented above, the average daily values of pollutants in raw sewage flowing to the septic tank were calculated, which are 705, 1400, and 650 mg $O_2$/dm$^3$ for BOD, COD, and TSS, respectively. The average, minimum, and maximum values of these parameters in the wastewater flowing out of the septic tank are presented in Table 1. The mean percentage elimination of pollutants in the septic tank was approximately 70% for BOD and COD and 86% for TSS, respectively. Large differences between the minimum and maximum concentrations are caused by the uneven inflow of sewage and pollutant loads. At certain times, the IES residents were absent, and, at other times, such as the start of the COVID-19 pandemic in 2020, there were twice as many permanent residents. Although the amount of sewage dosed to the pilot plant was constant throughout the entire period, the concentrations of pollutants underwent significant changes.

**Table 1.** Concentration of organic matter and suspended solids in the outflow of the septic tank.

| Value | Min | Max | Mean |
|---|---|---|---|
| Biological Oxygen Demand (BOD) mg $O_2$/dm$^3$ | 87 | 410 | 248.58 |
| Chemical Oxygen Demand (COD) mg $O_2$/dm$^3$ | 256 | 915 | 407.27 |
| Total Suspended Solids (TSS) dm$^3$ | 51.50 | 213.50 | 105.89 |

3.1.2. Elimination of Organic Matter and Suspended Solids in the Compost Beds

The average monthly concentrations in the outflow from the beds for the entire study period are presented in Figure 3a–c. Two stages can be distinguished in the outflow of compost beds. The first stage starts from the beginning of the study period and lasts until October 2020. During this time, a significant increase in BOD and COD concentrations is visible in the period from April to August. This increase was related, as mentioned above, to the time when more people lived in the IES building. This resulted in an increase in the concentration of raw sewage while maintaining a constant value of the daily hydraulic load of 70, 100, and 130 $l/m^2$ for the A, B, and C beds, respectively. Each of the beds reacted differently to this increase. The outflow from the first bed (sample 1A) shows the smallest change, which means that the compost bed with a daily hydraulic load of 70 $l/m^2$ is able to withstand periodic increases in concentration in the inflow and maintains stable values of the BOD and COD concentration in the outflow. On the other hand, the second bed (sample 1B) was less able to withstand the increase in concentration, which means that, in this case, the concentration in the beds outflow is influenced by the size of the hydraulic load. In this case, the increase in the BOD value in the outflow from this bed reached 100%. The third bed (sample 1C) with the highest hydraulic load reacted worst to the periodic increase in concentration. The increase in BOD concentration in the outflow was even above 200%. The overall characteristics of the amount of organic matter in the treated sewage and renewed water are presented in Table 2.

The second stage of the operation of the compost beds starts in October 2020 and lasted until the end of the measuring period. The water quality indicators describing the performance of compost beds in this period are presented in Table 3. During this time, the removal of organic matter expressed as BOD is very stable and is practically constant. Figure 3a shows that, despite the stability of the results, the highest concentration of BOD is maintained in the outflow from the third bed, which operated at the highest daily hydraulic load (130 $l/m^2$). Despite this difference, the concentrations in the outflow from all beds are still low and fluctuate around the value of 3 mg $O_2/dm^3$ (Table 3). On the other hand, the removal of organic matter expressed as COD is also more stable and, as can be seen from Figure 3b, there is a constant decrease in COD concentration in the outflow of the compost beds throughout the entire study period. This clearly indicates the constantly increasing efficiency of the elimination of organic matter expressed as COD in compost beds. In addition, Figure 3b shows the difference in concentrations that occur in the effluents of the A, B, and C beds. The lowest concentrations are seen in the outflow of the first bed (sample 1A), then the second bed (sample 1B), and then the third bed (sample 1C). However, in the end of the study period, these differences decreased significantly. Moreover, the effluent from the second bed showed lower COD concentrations than the effluent from the first bed. This may mean that, with the increasing working time of the beds, their efficiency in eliminating COD increases and the influence of the hydraulic load on the COD concentration in the effluent decreases. For the entire measurement period and for all beds, the average COD concentration in the outflow was 73 mg $O_2/dm^3$ (Table 2), whereas, from October 2020 until the end of the measuring period, the average COD value was 65 mg $O_2/dm^3$ (Table 3).

In the case of the elimination of suspended solids in the compost beds (Figure 3c), a strong stability of the concentrations in the effluent is visible, which, for all compost beds, oscillates around the value of 4 mg/$dm^3$ (Table 2). There is no visible influence of the hydraulic load on the concentration value in the outflow. This is confirmed by the fact that the values of the suspension concentration in the outflow from the first compost bed (sample 1A) exceed the values from the second and even the third compost bed (samples 1B and C). This means that, in the range of the tested hydraulic loads, compost beds stably remove the suspended solids from raw sewage. Only in the summer of 2020, when the concentration in the inflow to the beds was higher, was there a slight influence of the load on the TSS value in the outflow, but the differences are not as significant as in the case of BOD and COD.

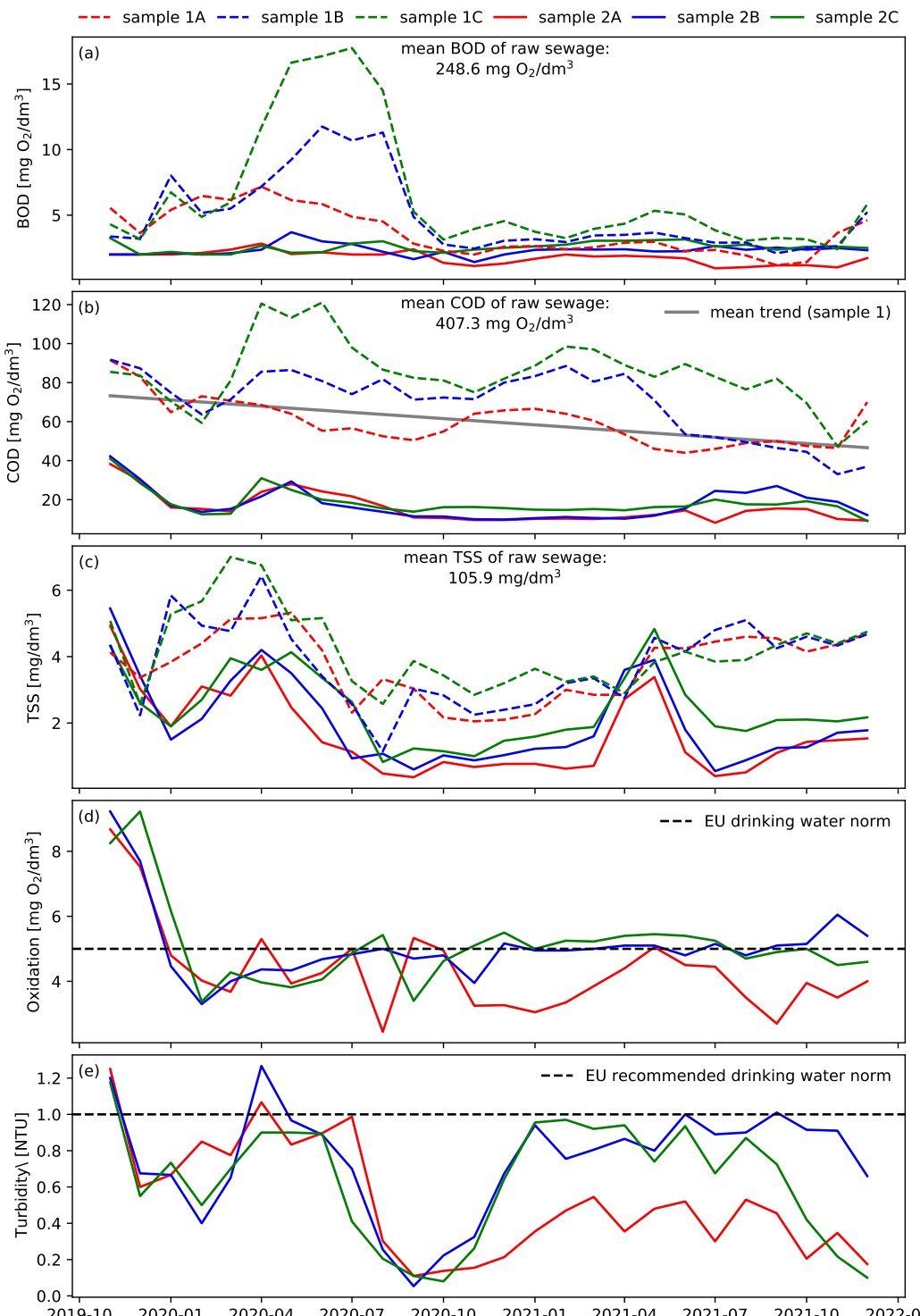

**Figure 3.** Quality of the water purified with the daily flow of 70 (A), 100 (B), and 130 l/m² (C), sampled after the compost bed (sample 1) and after the active carbon bed (sample 2). The following indicators are presented: (**a**) biological oxygen demand (BOD), (**b**) chemical oxygen demand (COD), (**c**) total suspended solids (TSS), (**d**) oxidation, and (**e**) turbidity (nephelometric turbidity unit—NTU). The drinking water quality recommendation is taken from the European Council Directive [22].

**Table 2.** Characteristics of treated wastewater (sample 1) and renewed water (sample 2) in the pilot plant.

| Value | Sample 1 | | | Sample 2 | | |
|---|---|---|---|---|---|---|
| | **Min** | **Max** | **Mean** | **Min** | **Max** | **Mean** |
| BOD (A) mg $O_2$/dm$^3$ | 1.14 | 8.90 | 4.08 | 0.52 | 4 | 1.81 |
| BOD (B) mg $O_2$/dm$^3$ | 1.25 | 12.68 | 5.46 | 0.80 | 4.20 | 2.30 |
| BOD (C) mg $O_2$/dm$^3$ | 1.50 | 19.68 | 7.23 | 1.32 | 5 | 2.52 |
| BOD total mg $O_2$/dm$^3$ | 1.14 | 19.68 | 5.59 | 0.52 | 5 | 2.21 |
| COD (A) mg $O_2$/dm$^3$ | 41 | 112 | 61.14 | 4 | 42 | 16.77 |
| COD (B) mg $O_2$/dm$^3$ | 29.60 | 111 | 72.04 | 9 | 56 | 17.90 |
| COD (C) mg $O_2$/dm$^3$ | 44.30 | 131 | 85.87 | 8 | 52 | 18.85 |
| COD total mg $O_2$/dm$^3$ | 29.6 | 131 | 73.02 | 4 | 56 | 17.84 |
| TSS (A) mg/dm$^3$ | 0.90 | 6.80 | 3.77 | 0 | 5.80 | 1.8 |
| TSS (B) mg/dm$^3$ | 0.50 | 7.40 | 3.87 | 0.10 | 6.80 | 2.13 |
| TSS (C) mg/dm$^3$ | 1.40 | 9.80 | 4.31 | 0.30 | 6.80 | 2.49 |
| TSS total mg/dm$^3$ | 0.50 | 9.80 | 3.98 | 0 | 6.80 | 2.14 |
| Oxidation (A) mg $O_2$/dm$^3$ | - | - | - | 1.40 | 10.20 | 4.51 |
| Oxidation (B) mg $O_2$/dm$^3$ | - | - | - | 2.50 | 12 | 5.10 |
| Oxidation (C) mg $O_2$/dm$^3$ | - | - | - | 2.50 | 12 | 5.16 |
| Oxidation total mg $O_2$/dm$^3$ | - | - | - | 1.40 | 12 | 4.92 |
| Turbidity (A) NTU | - | - | - | 0.02 | 2 | 0.55 |
| Turbidity (B) NTU | - | - | - | 0.01 | 2.10 | 0.71 |
| Turbidity (C) NTU | - | - | - | 0.01 | 2 | 0.61 |
| Turbidity total NTU | - | - | - | 0.01 | 2.10 | 0.63 |

**Table 3.** Concentration of organic matter and suspended solids after compost beds (sample 1) in the second stage of operation (from October 2020).

| Value | Min | Max | Mean |
|---|---|---|---|
| BOD (A) mg $O_2$/dm$^3$ | 1.14 | 4.70 | 2.56 |
| BOD (B) mg $O_2$/dm$^3$ | 1.25 | 5.78 | 3.10 |
| BOD (C) mg $O_2$/dm$^3$ | 2.22 | 6.57 | 3.89 |
| BOD total mg $O_2$/dm$^3$ | 1.14 | 6.57 | 3.18 |
| COD (A) mg $O_2$/dm$^3$ | 41 | 71.80 | 55.01 |
| COD (B) mg $O_2$/dm$^3$ | 29.60 | 89 | 63.39 |
| COD (C) mg $O_2$/dm$^3$ | 44.30 | 101 | 79.56 |
| COD total mg $O_2$/dm$^3$ | 29.60 | 101 | 65.99 |
| TSS (A) mg/dm$^3$ | 1.80 | 5.20 | 3.48 |
| TSS (B) mg/dm$^3$ | 2 | 5.50 | 3.70 |
| TSS (C) mg/dm$^3$ | 2.50 | 5.20 | 3.78 |
| TSS total mg/dm$^3$ | 1.80 | 5.20 | 3.65 |

Figure 4a shows the quarterly means of the percentage of organic matter and suspended solids elimination in the compost beds. The presented data show a very high average efficacy of 98%, 82%, and 96% for BOD, COD, and TSS removal, respectively. The difference between the effectiveness of BOD and COD elimination is due to the fact that the BOD parameter includes easily decomposing organic compounds, whereas the COD parameter includes organic matter that is difficult to decompose, which requires a much longer time for its mineralization. In addition, bacteria that break down such

contaminants require a longer adaptation period. This is visible in Figure 3b, where the trend of the decrease in COD concentration in the outflow from compost beds during the research is shown. This decrease means that, with the passage of time, the efficiency of the mineralization of these compounds in the bed by bacteria increases.

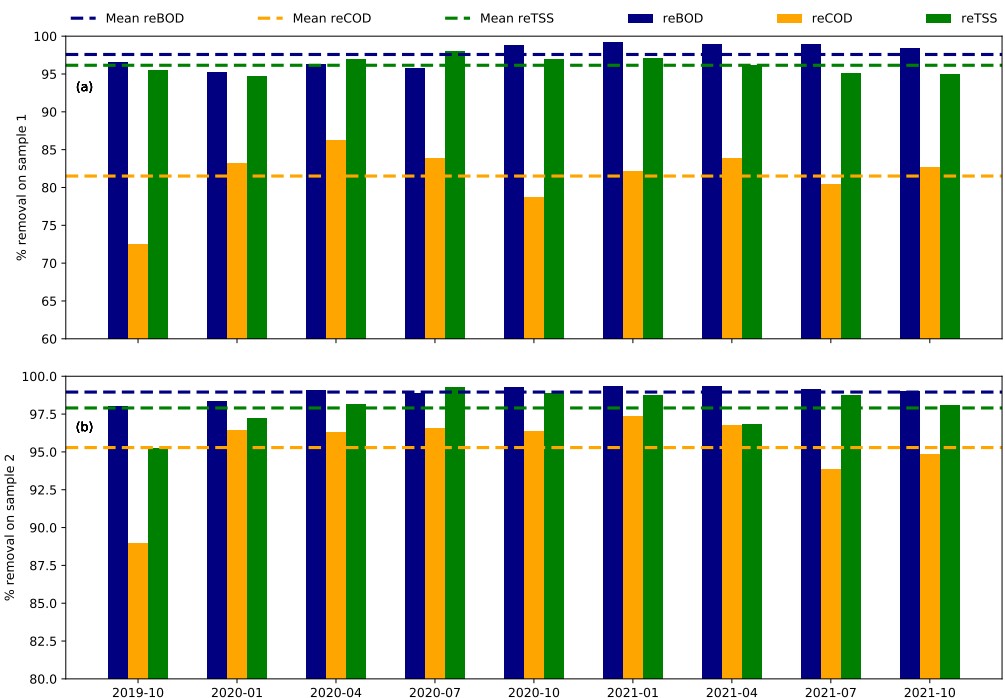

**Figure 4.** Percentage removal of BOD, COD, and TSS (reBOD, reCOD, and reTSS, respectively) in water sampled after the compost beds (**a**) and after the active carbon beds (**b**).

### 3.2. Elimination of Organic Matter and Suspended Solids in the Water Renewal Beds

In the pilot plant, the denitrification, phosphorus and active carbon beds perform the function of water renewal, i.e., they reduce the concentration of organic matter and suspended solids to the level allowed in drinking water. According to Polish [23] and EU [22] requirements, only organic substances expressed by oxidation are tested in drinking water. Its value should be less than $5 \, mg/dm^3$. BOD and COD are not regulated in drinking water requirements but are generally used to assess water quality. The amount of TSS is also not regulated by the drinking water requirement; it is only recommended that this amount be kept as low as possible. Often, instead of TSS, the water turbidity parameter is used, which is correlated with the TSS content in the water. Both Polish [23] and EU [22] recommendations require the NTU to be below 1.

The concentration of organic matter expressed as BOD is presented in Figure 3a, which shows that the water renewal beds, practically from the beginning of the research, are characterized by a very stable removal efficiency of this type of organic matter. Moreover, the value of BOD concentration in the outflow from the beds is independent of the BOD fluctuations in the inflow, which occurred especially in the first period of work. It is worth mentioning that the magnitude of the hydraulic load has no significant effect on the BOD concentration in the outflow of the beds. This is confirmed by the average values for individual beds for the entire measurement period, which oscillate around the value of $2 \, mg \, O_2/dm^3$ (Table 2).

Renewed water was also characterized by a high efficiency in the elimination of organic matter expressed as COD (Figure 3b). Moreover, there is no clear influence of the hydraulic load on the COD concentration in the effluent. This is confirmed by the average values of this concentration for individual water renewal beds for the entire measurement period, which oscillate around $17 \, mg \, O_2/dm^3$ (Table 2). In contrast to BOD, the concentration

of which in the outflow was practically the same throughout the entire measurement period, the concentration of COD in the outflow from the beds had a downward trend (Figure 3b). This confirms the earlier observations from compost beds that the effectiveness of removing difficult-to-decompose organic matter increases much more slowly, since the bacteria responsible for it need a much longer time to adapt to the specificity of difficult-to-decompose organic substances. In the final months of sampling, the COD concentration in the outflow from the water renewal beds decreased, on average, to the value of 10 mg/dm$^3$. It can be concluded that this is already the lower limit of the concentration of this type of organic substance in water and that the process of removing this substance from water has reached its maximum effectiveness. In this case, it can be concluded that it took 2 years for the bacteria mineralizing such organic substances to adapt to these substances so that they were completely eliminated from the water.

The concentration of TSS in the treated water is shown in Figure 3c. Although this figure seems to shows significant fluctuations in the concentration of suspended solids in the outflow, in fact, their values are also very stable, as this figure has a much smaller concentration range (from 0 to 6 mg/dm$^3$) than, e.g., the figure with BOD concentrations (from 0 to up to 20 mg/dm$^3$). In practice, as shown in Table 2, the average values of the suspension concentration in the outflow from the water renewal beds oscillate around the value of 2 mg/dm$^3$, which is very similar to the BOD concentration. This figure also shows a slight influence of the hydraulic load on the amount of suspension concentration in the outflow, but the differences are not great. The clearly lowest concentration was in the outflow from the first bed, but, as mentioned above, the difference in TSS concentrations from the second and third beds is practically negligible.

Figure 4b shows the quarterly means of the percentage of organic matter and suspended solids elimination in the water renewal beds. The water renewal process can be characterized with a very high average efficacy of 99%, 95%, and 98% for BOD, COD, and TSS removal, respectively. These values, however, refer to the elimination obtained in the compost and water renewal beds. Including the elimination ocurring in the septic tank, the overall removal efficiency of the pilot plant can be characterized by 99.69%, 98.73%, and 99.24% for BOD, COD, and TSS, respectively.

The variability in oxidation in the outflow of the water renewal beds is presented in Figure 3d, where the upper limit of acceptable oxidation (5 mg O$_2$/dm$^3$) in drinking water is also marked with a horizontal line [22]. From the beginning of the research period to the end of the first quarter, the value of oxidation decreased, then stabilized, and then, practically until the end of the measurement period, it oscillated within the limit for drinking water. Despite high values of oxidation in the outflow from the first bed (sample 2A) in the first quarter of the study, the average for the entire measurement period, amounting to 4.5 mg O$_2$/dm$^3$ (Table 2), meets the criteria of the maximum content of organic matter in drinking water. On the other hand, for the second bed (sample 2B), the average value of 5.1 mg O$_2$/dm$^3$ slightly exceeds the permissible value. A similar average value (5.16 mg O$_2$/dm$^3$) is obtained for the oxidation for the third bed (sample 2C). If the period of the beds' adaptation in the first quarter was omitted, then, in the remaining research period, the average oxidation concentration in the outflow from the second and third bed would also not exceed the permissible value. The relationship between the amount of hydraulic load and the oxidation concentration in the outflow from the second (sample 2B) and third (sample 2C) beds is small; practically non-existent. However, there is a small difference in relation to the first bed (sample 2A). In this case, the concentration in the outflow is clearly lower than the concentration in the second and third beds. Therefore, it gives a greater guarantee of maintaining the oxidation concentration in the outflow below the norm for drinking water.

The turbidity of the water in the outflow of the water renewal beds is presented in Figure 3e, where the upper limit of acceptable turbidity (1 NTU) in drinking water is also marked with a horizontal line [22]. Already in the middle of the first quarter of the study, the value of water turbidity was within the standard for drinking water. With the exception

of a short period at the end of the second quarter of the study, where the value of turbidity after the second bed (sample 2B) slightly exceeded the limit, in the remaining period, the value of turbidity in the outflows of all beds was within the standard for drinking water. On the other hand, the average values obtained for the entire measurement period, amounting to 0.5, 0.6, and 0.7 NTU in samples 2A, 2B, and 2C, respectively, are significantly below the norm (Table 2). Although there are visible differences in the value of turbidity in the outflow from individual beds, it is difficult to indicate the existence of a strong dependence of the value of turbidity on the hydraulic load. Especially in the first year of research, we can see in Figure 3e that, in several periods, the highest value of turbidity was indicated by the outflow from the first bed (sample 2A).

## 4. Discussion

### 4.1. Removal Efficiency in the Septic Tank

The efficiency of organic matter and suspended solids removal in septic tanks depends primarily on the ratio of the size of the septic tank to the amount of incoming sewage. This value determines the so-called hydraulic residence time. The longer the sewage remains in the tank, the greater the reduction in organic matter and suspended solids in it. In the studied pilot plant, the HRT in the septic tank was approximately 8 days and resulted in an average reduction of approximately 70% for BOD and COD and 86% for TSS. According to Halicki [24], in septic tanks with an HRT of 5 to 10 days, the reduction in organic matter expressed as BOD is 65 to 75%, and the reduction in TSS is approximately 95%. On the other hand, Colaves and Sandri [25] report that, under similar conditions, septic tanks provide an average of 65% for COD, 79% for BOD, and 87% for TSS. According to the research by Nasr and Haroun [26], septic tanks with an HRT of 3 days provided 65% for COD, 68% for BOD and 65% for TSS, respectively. In this case, it can be seen that the removal efficiency is lower due to the shorter retention time. Finally, it can be concluded that the results of the removal of organic matter and suspended solids in the septic tank obtained in our study confirm the efficiency of this method, as described in other studies.

### 4.2. Removal Efficiency in the Compost Beds

In this pilot plant, compost beds fulfill the function of treating raw sewage flowing out of the septic tank. The process of water renewal takes place in the next stage, in the denitrification, phosphorus, and activated carbon beds, where the purpose of which is to achieve the quality of drinking water. The compost beds have an area of 3 m$^2$, which is 25% of the NTSW pilot plant. All three compost beds (operating at different hydraulic loads) ensured a fairly stable and very effective elimination of organic matter and suspended solids (Table 2). When observing the average values of the concentration of organic matter, it is worth paying attention to the large COD / BOD ratio, which is as high as 14.6. In raw sewage, it is usually 2:1, and, in treated sewage, it increases to 4:1 or even a little more [27]. Taking into account the very low concentration of BOD in the outflow (5 mg $O_2/dm^3$) and the very high COD-to-BOD ratio, it can be concluded that compost beds remove practically all easily degradable organic matter, and only organic substances that are difficult to decompose, expressed as COD, remain in the treated sewage. This is also confirmed by the elimination percentage, which, for the entire period, was 98% for BOD and even 99% for the last 1.5 years. The relatively high value of the COD concentration in the outflow from the beds (73 mg $O_2/dm^3$ on average) is the effect of washing away humic substances from compost beds, which increased the COD concentration.

Compost beds, considered as a stage of wastewater treatment before its further renewal, ensured a very high efficiency of treatment. This is confirmed by the comparison of the concentrations achieved in the effluent to the values expected from treated wastewater. The values for municipal wastewater required by the EU directive are: 25 mg $O_2/dm^3$, 125 mg $O_2/dm^3$, and 35 mg /dm$^3$ for BOD, COD, and TSS, respectively [28]. Table 4 presents the Polish requirements for both municipal and domestic wastewater for sewage treatment plants of various sizes [29]. The quality of wastewater treated in compost beds

in terms of TSS and BOD removal is undoubtedly within the maximum values required by the EU directive and national regulations in Poland. Such a high degree of BOD and TSS treatment is possible in conventional municipal wastewater treatment plants only with the use of additional polishing processes, such as flotation and membrane filtration [30]. Any additional treatment processes in conventional treatment plants require additional energy consumption and increased costs. For example, the additional use of a membrane bioreactor increases the energy consumption by approximately 30% or by 1 kWh per cubic meter of treated sewage [31]. It should be emphasized that compost beds work without the need for an external energy supply. Wastewater is delivered to the beds by a pressure system, and the entire treatment process takes place spontaneously during the free filtration of wastewater through the compost bed.

**Table 4.** The highest permissible values of pollution indicators or the minimum percentages of pollution reduction for domestic and municipal wastewater treatment plants discharged into waters and soils in Poland [29]. N/A—not available.

| Parameter | Unit | Treatment Plant Size, Expressed by the Number of Inhabitants | | | |
|---|---|---|---|---|---|
| | | **<2000** | **2000–9999** | **10,000–14,999** | **15,000–99,999** |
| BOD | mg $O_2$/dm$^3$ (% reduction) | 40 (N/A) | 25 (70–90) | 25 (70–90) | 15 (90) |
| COD | mg $O_2$/dm$^3$ | 150 | 125 | 125 | 125 |
| TSS | mg /dm$^3$ | 50 | 35 | 35 | 35 |

*4.3. Removal Efficiency in the Water Renewal Beds*

The obtained results should first of all be assessed in terms of the main purpose of this study, i.e., meeting the drinking water quality standards, including the content of organic substances. According to the regulations in force in Poland [23] and the EU [22], drinking water should be characterized by oxidation lower than 5 mg $O_2$/dm$^3$ and turbidity below 1 NTU. The average values of oxidation and turbidity obtained in the pilot plant meet the criteria. COD and BOD are not standardized in these regulations, but the obtained values can be compared to the values of these parameters for clean waters. According to Chapman and Kimstach [32], in clean, uncontaminated surface waters, the BOD concentration fluctuates around 2 mg $O_2$/dm$^3$ and the COD is below 20 mg $O_2$/dm$^3$. This is confirmed by the regulations on the assessment of the quality of surface waters in Poland and the EU. According to these criteria, waters classified as streams and lowland rivers of the first quality class are characterized by a BOD concentration of $\leq$2.3 mg $O_2$/dm$^3$ [33]. This means that the recovered water from wastewater in the pilot installation meets the Polish and European criteria of the cleanest surface water in terms of its content of organic matter.

Due to its design, the use of natural treatment processes and the participation of plants, the pilot station can be classified as a constructed wetland or, generally, as a natural treatment system for wastewater. It should be added that the active area of the pilot treatment plant is only 3 m$^2$ per person, and the HRT (excluding the HRT of the septic tank) is approximately 6 days. The basic CW types, however, do not achieve this far removal of organic matter and suspended solids. Table 5 summarizes the average values that occur in the outflow of the three basic CW variants [34].

**Table 5.** Average values of the concentration of organic matter (BOD) and suspended solids (TSS) in the outflow of the three basic types of CW [34].

| CW Type | Hydraulic Residence Time (Days) | BOD (mg $O_2$/dm$^3$) | TSS (mg/dm$^3$) |
|---|---|---|---|
| Free water surface | 7–15 | 5–10 | 5–15 |
| Subsurface flow | 3–14 | 5–40 | 5–20 |
| Vertical flow | 1–2 | 5–10 | 5–10 |

The practical application of combined systems, i.e., consisting of various types of CW as well as other types of natural treatment systems, allows us to achieve a similar effect

as in the case of the pilot plant. For example, such a multistage treatment plant consisting of a number of different CW and treatment ponds operating in Norway achieved similar results in terms of the elimination of BOD and TSS [3]. It is worth mentioning that its active area per person is 10 m$^2$, and the HRT in the treatment plant is 75 days, which definitely distinguishes it from the described pilot plant. It should be noted that, in our study, the low values of BOD and TSS concentration were achieved in the pilot plant practically after compost beds, which occupy an active surface of $\leq$1 m$^2$ per person with an HRT of approximately one hour.

### 4.4. Possibility of Reusing the Reclaimed Water

The main goal of the project, within which the pilot plant was built, was to treat wastewater to the quality of drinking water. The purified water was to be safely reused for flushing toilets, the irrigation of home gardens, greenery, and other environmental purposes, such as creating new habitats for fauna and flora or supplying groundwater. The comparison of the water parameters achieved in the pilot plant with the requirements of water recovered from wastewater and intended for reuse in Europe [4], Canada [35], and USA [36] is presented in Table 6. It should be clarified that the given European requirements apply to water intended for watering crops that are directly intended for consumption. They are therefore the highest water quality criteria used in Europe. The American requirements also apply to the highest category of water quality requirements and they concern the reuse of water for the direct irrigation of crops and the use of water for non-drinking purposes in municipal management. In contrast, Canadian requirements apply to the reuse of renewed water for flushing toilets.

**Table 6.** Comparison of the quality of water treated in the pilot plant to European [4], Canadian [35], and American [36] quality requirements for water intended for reuse. N/A—not available.

| Parameter | Pilot Plant | EU | USA | Canada |
|---|---|---|---|---|
| BOD (mg $O_2$/dm$^3$) | 2 | $\leq$10 | $\leq$10 | $\leq$10 |
| TSS (mg/dm$^3$) | 2 | $\leq$10 | N/A | $\leq$10 |
| Turbidity (NTU) | 0.6 | $\leq$5 | $\leq$2 | $\leq$2 |

As shown in Table 6, the quality of water from the pilot plant in the scope of BOD, TSS, and turbidity meets the standards of water intended for reuse. This means that water treated in the pilot plant should not pose a threat to the health of residents who may have direct or indirect contact with such water. This is important because the use of renewed water from wastewater for irrigation purposes still raises many uncertainties in relation to the remaining micro-pollutants in the water after the treatment process, such as pharmaceuticals, which pose certain threats to human health [5,37]. In addition, attention should be paid to the fact that the reuse of renewed water used so far concerns the outflows from large treatment plants. They are intended for irrigation, mainly in dry areas (e.g., Spain, Israel, California, Australia), where water can be used all year round. In Poland and other countries of Central and Western Europe, agricultural use may be mainly seasonal, which significantly reduces the possibility of reusing all of the water recovered from wastewater. The presented research results indicate that small home sewage treatment plants can guarantee even a much better quality of water recovered from sewage than that required by the standards. On the other hand, the local use of this water for flushing toilets and watering green areas can save approximately 50% of drinking water taken from the water supply network throughout the year. In the autumn and winter period, when there is no need for watering, water of such good quality is suitable for supplying groundwater. Therefore, all of the renewed water can be utilized. However, to obtain such an effect, the additional construction of a dual pipe distribution system is required. An example of reusing purified water (using the dual pipe distribution system) is the city of St. Petersburg (Florida, USA), where the renewed water reduced the city's clean water needs by 40% [38].

The described pilot treatment plant with water renewal allows not only to reuse the renewed water. It also enables the reuse of many macro and microelements that accumulate in the compost bed during the purification process. As mentioned earlier, the compost bed consists of 80% wood chips, which during a few years of cleaning process turns into a compost rich in micro and macro elements, which can be used for fertilization in home gardens. It is important not only to use macro and micronutrients, but also the compost itself as an organic fertilizer, which has a very positive effect on the improvement of soil properties. Thus, the described example of a sewage treatment plant allows for a very comprehensive way to reuse water and at the same time to produce valuable compost.

## 5. Conclusions

This study presents the results of an almost 2.5-year operation of a natural treatment system for wastewater constructed by the building of the Institute of Applied Ecology in Western Poland, in which, there were three regular inhabitants and up to five employees. The pilot plant consisted of a septic tank, compost beds, and denitrification, phosphorus, and activated carbon beds. The active area of the pilot plant was only 3 m$^2$ per person, and the HRT (excluding the septic tank with HRT of 8 days) was 6 days. Wastewater treatment was carried out in compost beds, while in the remaining beds water renewal took place. The obtained results show a very high removal efficiency of the pilot plant (99%, 96%, and 98% for BOD, COD, and TSS, respectively). The renewed water was characterized by average concentrations of 2.2 mg $O_2$/dm$^3$, 17.8 mg $O_2$/dm$^3$, 2.1 mg/dm$^3$, 4.9 mg $O_2$/dm$^3$, and 0.6 NTU for BOD, COD, TSS, oxidation, and turbidity, respectively. Therefore, this water met the Polish and European drinking water criteria, in which, the oxidation and turbidity thresholds are 5 mg $O_2$/dm$^3$ and 1 NTU, respectively. Water renewed in this pilot plant is used for toilet flushing and irrigation. This can allow for up to 50% savings of water supply; therefore, this installation can be especially attractive in dry regions with limited water resources.

**Author Contributions:** Conceptualization, W.H.; methodology, W.H.; software, M.H.; validation, W.H.; formal analysis, M.H.; investigation, W.H.; resources, W.H.; data curation, M.H.; writing—original draft preparation, W.H. and M.H.; writing—review and editing, W.H. and M.H.; visualization, M.H.; supervision, W.H.; project administration, W.H.; funding acquisition, W.H. All authors have read and agreed to the published version of the manuscript.

**Funding:** This research was funded by National Centre for Research and Development grant number POIR.01.01.01-00-0805/18.

**Conflicts of Interest:** The authors declare no conflict of interest.

## Abbreviations

The following abbreviations are used in this manuscript:

| | |
|---|---|
| BOD | Biological Oxygen Demand |
| COD | Chemical Oxygen Demand |
| CW | Constructed Wetland |
| IES | Instytut Ekologii Stosowanej (Institute of Applied Ecology) |
| HRT | Hydraulic Residence Time |
| NTSW | Natural Treatment System for Wastewater |
| NTU | Nephelometric Turbidity Unit |
| TSS | Total Suspended Solids |

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
