# Peer review of "From Domestic Sewage to Potable Water Quality: New Approach in Organic Matter Removal Using Natural Treatment Systems for Wastewater"

_water, doi:10.3390/w14121909_

Round 1
Reviewer 1 Report
The title of the manuscript indicates that sewage will be treated to Potable water quality, however the whole manuscript discusses the possibility to treat wastewater for reuse in other ways but not for drinking water, therefore the title is misleading.
In the manuscript there are relevant part where appropriate citation is needed such as: Line 24-26, 143-156 etc.
The authors use Compost filter treatment method but is not properly described using scientific literature references. In my opinion the authors should change the name of the treatment step to “filter bed”, the compost filter name is misleading because in other environmental technologies compost filter is used to eliminate odour generated during composting.
There are lot of unclear sentences, which can’t be understood correctly such as:
Line 37: “…of which is directed to irrigation in agriculture”
Line 64-65: ”…NTSW pilot plant built by the building of the Institute of Applied Ecology…”
Line 118: “…The renewed water reservoir is used…”
Line 130: Title 2.3 “Water purification process” – not domestic wastewater is treated?
Line 167: “….process stage of water renewal, which takes place during…” – incorrect naming of the treatment “renewal”
Line 226-227: “….the operation of the compost filter starts in October 2020 and lasts…” (incorrect past tense use)
Etc…
Other recommendations:
Line 120: Subtitle 2.2 Water sampling and quality assessment – the authors should use correct references; it is advised to use methods form APHA Standard Methods for Water and Wastewater Analysis.
Also, the authors have to clarify the difference between COD and Oxidation measurements because for the reader is very confusing, why Permanganate method is also applied (in principle de COD method is more efficient in oxidation capacity point of view of the sample).
Subtitle 2.3. third paragraph the authors indicated that the effluent (outflow) from the denitrification plant had humic substances which can be attributed to the use of so-called compost filter (peat soil – has high humic content) and also from the denitrification bed composition (swamp sediment) through the water washes out the humic substances.
In the Materials and methods section paragraph between lines 96-104 the authors state the denitrification bed is not involved in removal of organic matter which is an incorrect statement because trough denitrification process facultative heterotrophic bacteria to reduce nitrate use az electron donor organic matter.
Figure 4. should be remade its unclear to see the results.
Etc.
The whole manuscript has to be proof read, the misleading wording of sentences need to be corrected and a more clear and correct presentation is advised.
Author Response
- Request to change the title. We agree with the Reviewer, that the title might be misleading. We did not intend to reuse the treated water as drinking water, however we aimed to achieve drinking water quality, which has been shown in the discussion. Therefore, we changed the frst part of the title to "From Domestic Sewage to Potable Water Quality".
- Request to add citation in lines 24-26. In spite of adding citation, we showed, that the number of articles available in the Science Direct database (https://www.sciencedirect.com/, accessed on 22.05.2022) related with the "constructed wetland", "water reuse" and "water renewal" keywords increased significantly over the past two decades, amounting to 324, 1889, and 706 in 2000 and 3139, 23 228, and 4129 in 2020, respectively.
- Request to add citation in lines 143-156. We agree that this paragraph needs some more references. Therefore, we added two articles in this paragraph.
- Request to change the naming of the treatment step from compost filter to "filter bed". We thank the Reviewer for this remark. However, the "compost filter" term is a translation of the original name of this part of the treatment plant. This name appears both in the description of the project (by which this study was financed) and in the descriptions of technical solutions on the basis of which the pilot treatment plant was built. In addition, the novelty of this solution is the use of the compost environment for the purification process. On a compromise basis, we can replace the "compost filter" with "compost bed", since the denitrification, phosphorus and carbon beds are also called beds. It would be more uniform that way. However, since in the manuscript the denitrification, phosphorus and activated carbon beds were sometimes called "beds", we changed these occurences to "water renewal beds".
- Line 37: Request to change the unclear sentence "of which is directed to irrigation in agriculture". We split this sentence into two: "In Spain, which in Europe is the leader in this respect, about 11% of treated wastewater is reused. 62% of the reused water is directed to irrigation in agriculture."
- Lines 64-65: Request to change the unclear sentence "NTSW pilot plant built by the building of the Institute of Applied Ecology". We changed this sentence to: "This study aims to describe the performance of a NTSW pilot plant operating by the building of the Institute of Applied Ecology (Instytut Ekologii Stosowanej - IES) in W Poland."
- Line 118: Request to change the unclear sentence "The renewed water reservoir is used". We changed this sentence to: "The renewed water reservoir (with a capacity of 5 m3) serves as a store of water recovered from sewage."
- Line 130: Request to change the sentence "Water purification process". We thank the Reviewer for this remark. The corrected sentence is "Wastewater purification process".
- Line 167: Request to change the incorrect naming of the treatment "renewal". We do not understand, why is "water renewal" an incorrect term. However, we simplified this sentence to: "The water renewal process takes place in the denitrification, phosphorus and activated carbon beds."
- Line 226: Request to change the tense in the sentence "the operation of the compost filter starts in October 2020 and lasts". We thank the Reviewer for this remark. We changed the sentence to: "The second stage of the operation of the compost beds started in October 2020 and lasted until the end of the measuring period."
- Line 120: Request to use methods from APHA Standards Methods for Water and Wastewater Analysis. We thank the Reviewer for this remark. We updated the whole paragraph and used the methods suggested by the Reviewer.
- Request to clarify the difference between COD and Oxidation. We added a description at the end of the 2.2. subsection.
- Lines 64-65: Request to clarify the participation of the denitrification bed in the process of organic matter removal. In our opinion, the presented description is correct, because the processes of removing nitrates in the denitrification bed have no effect on the removal of organic matter remaining in the treated sewage (after compost beds) that flow through the denitrification bed. The reviewer is right that the bacteria that reduce nitrates also oxidize the organic substance. However, as noted in the description, the denitrification bed is filled with a 40 cm layer of organic sediments and these are the source of organic matter for denitrifying bacteria. Denitrifying bacteria prefer to oxidize easily degradable matter, such as the one mentioned above. On the other hand, the inflow from compost filters contains organic substances that are difficult to decompose and only a very small amount of easily decomposable substances, as evidenced by the COD / BOD ratio, which on average amounts to 14.6 (description in section 4.2). Therefore, we suggest simply refining the sentence "The bed is not involved in the removal of organic matter contained in the treated water" (lines 108-109).
- Request to remake the Figure 4, as it appears unclear. We added more descriptive text to the figure caption. Further, we also changed the colours, so they are more distinguishable. We also increased the size of the dashed lines, so they can be better perceived.
- Also, following the general request to proof read the manuscript, we consulted the whole document with an english native speaker. Therefore some aditional small changes occur in the manuscript.
Reviewer 2 Report
Dear Editor: This paper has proposed good idea to recycling the wastewater by a natural treatment system. It is worth to be published here. But more data is requested to proof the change in pH of recycled water.
Author Response
- Request to present more data on the change in pH of the recycled wa-
ter. We agree with the Reviewer that this data should be presented. Therefore,
we added two sentences starting at the line 118.
Reviewer 3 Report
The research work presented is well conducted and presented in a scientific manner. The work explains the water treatment methodology at pilot level, which includes a case study. The manuscript can be accepted after minor revision and include the given comments.
1. The authors did extensive literature studies on wastewater treatment and its reuse for various purposes mainly irrigation of agricultural land. It's better to represent the important data in a comparison table (includes country, technique, and percent usage of wastewater and other parameters).
- Also, include a brief discussion on the treatment of recovered waste after wastewater treatment.
Author Response
- Request to present the data on wastewater treatment and reuse in
different countries in tables. We thank the the Reviewer for this comment.
However, we believe that it would be difficult to tabulate the amount of water reuse using different techniques and methods in different countries. One of the reasons for this is that the qualities of water that are reused are very different. Many countries use normal runoff from treatment plants for reuse in agriculture, others use different quality categories for reuse water. In order to objectively reflect the actual state of water reuse, much more data should be provided than can be done in the table. For this reason, the authors suggest leaving the description of water reuse in various countries in a simplified form as in the text. - Request to include a brief discussion on the treatment of recovered waste after wastewater treatment. We agree with the Reviewer that this data should be presented. Therefore, we added a short paragraph at the end of the 4.4. subsection.